# Dynamic Modeling of a Front-Loading Type Washing Machine and Model Reliability Investigation

**Jungjoon Park [1], Sinwoo Jeong [2,\*] and Honghee Yoo [3,\*]**

[1] Samsung Electronics, Suwon 16677, Korea; jungjoonp93@gmail.com
[2] Department of Mechanical Convergence Engineering, Hanyang University, Seoul 04763, Korea
[3] Department of Mechanical Engineering, Hanyang University, Seoul 04763, Korea
\* Correspondence: koreawoo11@hanyang.ac.kr (S.J.); hhyoo@hanyang.ac.kr (H.Y.);
Tel.: +82-2-2220-0446 (S.J. & H.Y.)

**Abstract:** A linear dynamic model of a front-loading type washing machine was developed in this study. The machine was conceptualized with three moving rigid bodies, revolute joints, springs, and dampers along with prescribed rotational drum motion. Kane's method was employed for deriving the equations of motion of the idealized washing machine. Since the modal and transient characteristics can be conveniently investigated with a linear dynamic model, the linear model can be effectively used for the design of an FL type washing machine. Despite the convenience, however, the reliability of the linear dynamic model is often restricted to a certain range of system parameters. Parameters relevant to the reliability of the linear dynamic model were identified and the parameters' ranges that could guarantee the reliability of the proposed linear dynamic model were numerically investigated in this study.

**Keywords:** washing machine; front-loading type; dynamic modeling; modal characteristic; transient characteristic; model reliability



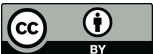

## 1. Introduction

Most washing machines manufactured today consist of a cabinet, tub, motor, drum, and suspension that consists of springs and dampers. The drum is usually embraced by a tub and it rotates around an axis that is either horizontal or vertical to the base plane. If the rotating axis is vertical to the base plane, the washing machine is classified as the top-loading type; if the rotating axis is horizontal to the base plane, it is classified as the front-loading type (hereafter, FL type). Compared to the top-loading type, the FL type generally uses less water and detergent and is known to cause less damage to clothes. However, since the FL type initially had relatively limited washing capacity, it was mostly used in Europe where water is expensive and hard to obtain [1]. Usage of the FL type has increased recently because its washing capacity has significantly increased. However, since the FL type is generally heavier than the top-loading type, its energy efficiency has not been satisfactory. In order to enhance its energy efficiency, its weight has been continuously reduced, but this has resulted in the serious problem of harsher vibrations [2].

Compared to the top-loading type, the FL type is, by the nature of its configuration, more likely to face problems such as rotating mass unbalance and machine walking. Such problems mostly occur during the machine's fast-spin cycle and are caused by unevenly distributed clothes attached to the inside surface of the drum. During wash and rinse cycles, the operation speed is relatively slow such that significant vibration hardly occurs. During the dehydration cycle at high operation speeds, however, clothes unevenly attach to the inside surface of the drum and create rotating mass unbalance. This unbalance often induces excessive vibration, which may result in various undesirable phenomena such as unpleasant noise, machine walking, and structural failures of the machine components.

Many studies have been conducted in order to analyze the dynamic behavior of the FL type washing machine exhibiting rotating mass unbalance. These machines usually have friction dampers and rubber bushings that have nonlinear dynamic characteristics. For designing such washing machines, it is hard to develop a reliable computational model to predict their dynamic behavior. Previous studies mostly used simplified numerical models for the dynamic analysis of an FL type washing machine. Conrad and Soedel [3] made a simple 2-D model of an FL type washing machine in order to investigate machine walking phenomena. Papadopoulos and Papadimitrious [4] proposed a simple 3-D model of a portable FL type washing machine and improved its dynamic stability. They provided expert insight on how an analytical model of an FL type washing machine could be developed and used for the dynamic analysis, design, and control of such machines. However, the effects of the machine's springs and dampers were not considered in their model.

Boyraz and Gündüz [5] proposed a 2-D model of an FL type washing machine including springs and dampers and conducted an optimization to minimize the vibration amplitudes of the tub. Buśkiewicz et al. [6] proposed a 3-D model of an FL type washing machine that included a rotating drum and a non-rotating tub. The drum rotated inside the tub but other relative movements between the tub and drum were not considered; thus, possible clashing between the tub and the drum under excessive vibration could not be predicted with the model. Lim et al. [7] proposed a 3-D model of an FL type washing machine that included additional degrees of freedom to consider the effect of elastic deformation between the tub and the drum. They also considered the bearings between the tub and the shaft as stiffness elements in the radial direction; thus, the clashing phenomena could be predicted with their model. Kamarudin et al. [8] derived linear equations of motion of an FL type washing machine and conducted dynamic, frequency response, and relative movement analyses between the tub and drum. The linear equations of motion were useful for the modal and frequency response analyses and control system design, but in some cases the dynamic behavior of the machine could not be accurately predicted with the linear dynamic model. In other words, the limitations and reliability of the linear dynamic model were not investigated in their work.

In this study, a washing machine was modeled as a multibody system posessing three rigid bodies, multiple joints, springs, and dampers along with a prescribed rotational drum motion. The equations of motion were derived using Kane's method [9]. The integrity of the three rigid body model was previously well addressed in [6–8]. The accuracy of the nonlinear analytical three rigid body model was first validated by comparing its numerical results to those obtained with a commercial multibody dynamic analysis software [10]. Then, by linearizing the nonlinear equations, the linear equations of motion were obtained. By using the linear analytical model, modal and transient analyses were conducted to investigate the dynamic characteristics of the machine. Moreover, the reliability of the linear analytical model was examined by comparing its transient results to those of the fully nonlinear model. By a comparison study, the reliability and limitations of the linear analytical model suggested in this study could be investigated.

## 2. Modeling an FL-Type Washing Machine

### 2.1. System Configuration

Figure 1 shows a schematic of the washing machine model. In the multibody model, body T is the tub that is supported by four springs and two dampers; body S is the shaft and rotor; and body D is the drum to which an unbalance mass is attached. Body T is connected to the cabinet by four springs and two dampers, and the cabinet is assumed to be fixed to the ground. In order to simplify modeling, the springs are idealized as linear springs and the dampers are idealized as linear viscous dampers. Body S and body T are connected with two bearings that are idealized as linear springs in the radial direction. Body D is attached to the end of the shaft of body S. The bending stiffness between the two bodies (D and S) is defined to consider the lateral elastic deformation of the drum-shaft assembly. The effect of the stiffness model on the dynamic behavior of an FL type washing

machine was discussed in previous studies [7,8]. In this study, the rotational motion of the drum during the dehydration cycle was prescribed as a function of time.

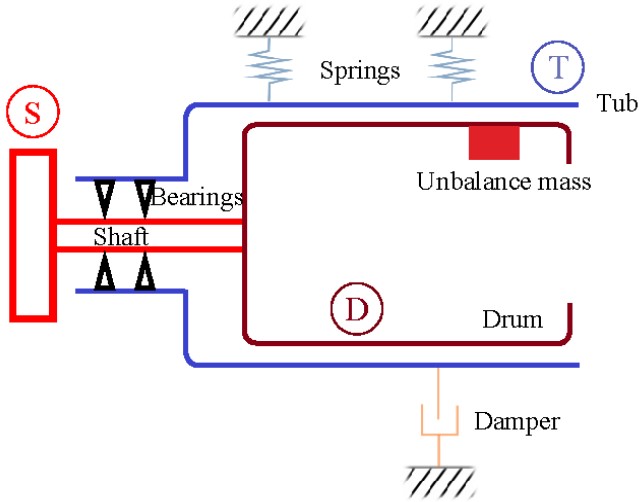

**Figure 1.** The schematic of a simplified front-loading type washing machine model.

### 2.2. Derivation of Equations of Motion

Using Kane's method, the equations of motion of the idealized FL washing machine model shown in Figure 1 were derived based on the following assumptions:

1. The cabinet is fixed to the ground.
2. Axial rotational motion of the tub is ignored.
3. Torsional deformation of the shaft relative to the tub is ignored.
4. Translational motion of the drum relative to the tub in the axial direction is ignored.
5. An unbalance mass is fixed to the drum.
6. The rotational motion of the drum is prescribed as a function of time.

Figure 2 shows the side and front views of the assemblies. In Figure 2a, $P_T$ is the midpoint between the front and rear bearing centers. $M_T$ is the mass center of body T. Lengths $L_{fb}$, $L_{rb}$, and $L_T$ denote the distances of the front bearing center, rear bearing center, and mass center from $P_T$, respectively. $F_{fb}$ and $F_{rb}$ denote the forces acting on body T exerted by the front and rear bearings, respectively. The coordinate system $\hat{c}_i$'s is attached to the cabinet C, which is assumed to be fixed in space, and point $O$ is the reference point fixed to the ground. The displacements of $P_T$ relative to $O$ in the directions of $\hat{c}_2$ and $\hat{c}_3$ are represented by two generalized coordinates, $q_1$ and $q_2$. The side and front views of the drum-shaft assembly are shown in Figure 2b. $P_S$ is the midpoint between the front and rear bearing centers in S. $M_S$ and $M_D$ are the mass centers of body S and body D, respectively. $P_D$ is the center of the drum rear side, and $P_{DE}$ is the center of the drum front side. Lengths $L_S$ and $L_D$ are the distances of $M_S$ and $P_D$ from $P_S$, respectively. $L_{DM}$ and $L_{DE}$ are the distances of $M_D$ and $P_{DE}$ from $P_D$, respectively. The mass center $M_D$ is shifted away from the central axis of the drum by $e$ due to the unbalanced mass attached to the drum. $\hat{c}_2$ and $\hat{c}_3$ measure numbers of the displacement of $P_S$ relative to $O$ and are represented by two generalized coordinates $q_3$ and $q_4$.

Figure 3 shows the body 2-3 rotations of the three rigid bodies. The coordinate systems $\hat{t}_i$ $(i = 1, 2, 3)$, $\hat{s}_i$ $(i = 1, 2, 3)$, and $\hat{d}_i$ $(i = 1, 2, 3)$ are fixed to body T, body S, and body D, respectively. Each body has 2 degrees of freedom for the rotational motion. The generalized coordinates $\hat{q}_i$ $(i = 5, 6, 7, 8)$ represent the Euler angles of body 2-3 rotations of body T and body S relative to the cabinet. Generalized coordinates $\hat{q}_i$ $(i = 9, 10)$ represent the Euler angles of body 2-3 rotation of body D relative to body S. A prescribed motion $\theta(t)$ is given to body S in the direction of the shaft axis $\hat{s}_1$. Table 1 shows the direction cosine tables among the four coordinate systems.

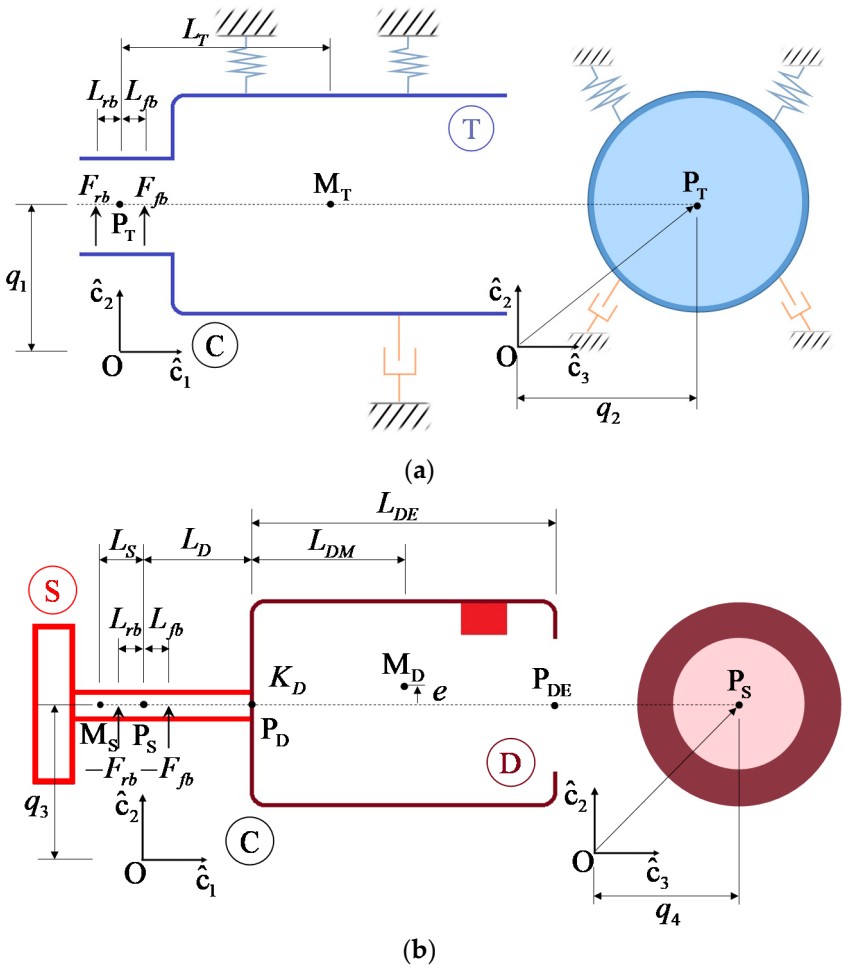

**Figure 2.** Side and front views of the assemblies: (**a**) tub assembly and (**b**) drum-shaft assembly.

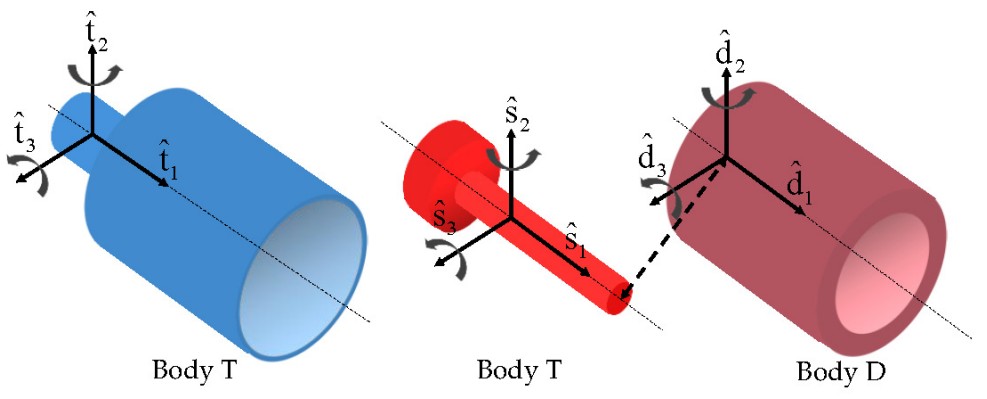

**Figure 3.** Body rotations and corresponding generalized coordinates of the three rigid bodies.

**Table 1.** Direction cosine tables among the four coordinate systems.

|  | $\hat{\mathbf{t}}_1$ | $\hat{\mathbf{t}}_2$ | $\hat{\mathbf{t}}_3$ |  | $\hat{\mathbf{d}}_1$ | $\hat{\mathbf{d}}_2$ | $\hat{\mathbf{d}}_3$ |  | $\hat{\mathbf{s}}_1$ | $\hat{\mathbf{s}}_2$ | $\hat{\mathbf{s}}_3$ |
|---|---|---|---|---|---|---|---|---|---|---|---|
| $\hat{\mathbf{c}}_1$ | $c_5 c_6$ | $-c_5 s_6$ | $s_5$ | $\hat{\mathbf{s}}_1$ | $c_9 c_{10}$ | $-c_9 s_{10}$ | $s_9$ | $\hat{\mathbf{c}}_1$ | $c_7 c_8$ | $s_7 \sin\theta - c_7 s_8 \cos\theta$ | $c_7 s_8 \sin\theta + s_7 \cos\theta$ |
| $\hat{\mathbf{c}}_2$ | $s_6$ | $c_6$ | $0$ | $\hat{\mathbf{s}}_2$ | $s_{10}$ | $c_{10}$ | $0$ | $\hat{\mathbf{c}}_2$ | $s_8$ | $c_8 \cos\theta$ | $-c_8 \sin\theta$ |
| $\hat{\mathbf{c}}_3$ | $-s_5 c_6$ | $s_5 s_6$ | $c_5$ | $\hat{\mathbf{s}}_3$ | $-s_9 c_{10}$ | $s_9 s_{10}$ | $c_9$ | $\hat{\mathbf{c}}_3$ | $-s_7 c_8$ | $c_7 \sin\theta + s_7 s_8 \cos\theta$ | $-s_7 s_8 \sin\theta + c_7 \cos\theta$ |

In order to derive the equations of motion using Kane's method, velocities of the mass center points and the angular velocities of the three rigid bodies need to be expressed in terms of generalized speeds. The following simplest form of generalized speeds is employed in this study.

$$u_i = \dot{q}_i \quad (i = 1, 2, \cdots, 10) \tag{1}$$

Now, the velocities of points $P_T$ and $P_S$ are expressed as follows.

$$\vec{v}^{P_T} = u_1 \hat{c}_2 + u_2 \hat{c}_3 \tag{2}$$

$$\vec{v}^{P_S} = u_3 \hat{c}_2 + u_4 \hat{c}_3 \tag{3}$$

The angular velocities of the three rigid bodies can be expressed as follows:

$$\vec{\omega}^T = u_5 s_6 \hat{t}_1 + u_5 c_6 \hat{t}_2 + u_6 \hat{t}_3 \tag{4}$$

$$\vec{\omega}^S = (\dot{\theta} + u_7 s_8)\hat{s}_1 + (\cos\theta c_8 u_7 + \sin\theta u_8)\hat{s}_2 + (-\sin\theta c_8 u_7 + \cos\theta u_8)\hat{s}_3 \tag{5}$$

$$\vec{\omega}^D = \vec{\omega}^S + {}^{S}\vec{\omega}^D \tag{6}$$

where the angular velocity of D in S can be expressed as follows.

$$^{S}\vec{\omega}^D = u_9 s_{10} \hat{d}_1 + u_9 c_{10} \hat{d}_2 + u_{10} \hat{d}_3 \tag{7}$$

In Equations (4), (5) and (7), and Table 1, $s_i$ denotes $\sin\theta_i$, and $c_i$ denotes $\cos\theta_i$.

Now, the velocities of the three mass centers $M_T$, $M_S$, and $M_D$ can be obtained using the following formulas:

$$\vec{v}^{M_T} = \vec{v}^{P_T} + \vec{\omega}^T \times L_T \hat{t}_1 \tag{8}$$

$$\vec{v}^{M_S} = \vec{v}^{P_S} + \vec{\omega}^S \times (-L_S)\hat{s}_1 \tag{9}$$

$$\vec{v}^{M_S} = \vec{v}^{P_S} + \vec{\omega}^S \times (-L_S)\hat{s}_1 \tag{10}$$

where the following is the case.

$$\vec{v}^{P_D} = \vec{v}^{P_S} + \vec{\omega}^S \times L_D \hat{s}_1 \tag{11}$$

In order to derive the generalized inertia forces, the angular accelerations of the three rigid bodies and accelerations of the three mass centers need to be obtained using the following formulas.

$$\vec{\alpha}^T = \frac{d\vec{\omega}^T}{dt} \tag{12}$$

$$\vec{a}^{M_T} = \vec{a}^{P_T} + \vec{\omega}^T \times (\vec{\omega}^T \times L_T \hat{t}_1) + \vec{\alpha}^T \times L_T \hat{t}_1 \tag{13}$$

$$\vec{\alpha}^S = \frac{d\vec{\omega}^S}{dt} \tag{14}$$

$$\vec{a}^{M_S} = \vec{a}^{P_S} + \vec{\omega}^S \times (\vec{\omega}^S \times (-L_S)\hat{s}_1) + \vec{\alpha}^S \times (-L_S)\hat{s}_1 \tag{15}$$

$$\vec{a}^{P_D} = \vec{a}^{P_S} + \vec{\omega}^S \times (\vec{\omega}^S \times L_D \hat{s}_1) + \vec{\alpha}^S \times L_D \hat{s}_1 \tag{16}$$

$$\vec{\alpha}^D = \frac{d\vec{\omega}^D}{dt} \tag{17}$$

$$\vec{a}^{M_D} = \vec{a}^{P_D} + \vec{\omega}^D \times (\vec{\omega}^D \times L_{DM}\hat{d}_1) + \vec{\alpha}^D \times L_{DM}\hat{d}_1 \tag{18}$$

If the velocity of the mass center of the $k^{\text{th}}$ rigid body is described as $\overset{\rightarrow k}{v}$ and the angular velocity of the $k^{\text{th}}$ rigid body is described as $\overset{\rightarrow k}{\omega}$, the partial velocity of the $k^{\text{th}}$ rigid body mass center $\overset{\rightarrow k}{v}_r$ and the partial angular velocity of the $k^{\text{th}}$ rigid body $\overset{\rightarrow k}{\omega}_r$ are obtained as follows.

$$\overset{\rightarrow k}{v}_r = \frac{\partial \overset{\rightarrow k}{v}}{\partial u_r} \tag{19}$$

$$\overset{\rightarrow k}{\omega}_r = \frac{\partial \overset{\rightarrow k}{\omega}}{\partial u_r} \tag{20}$$

Then, the generalized inertia force $F_r^*$ can be obtained by using inertia force $\overset{\rightarrow *}{F}_k$ and inertia torque $\overset{\rightarrow *}{T}_k$ of the $k^{\text{th}}$ rigid body as follows:

$$F_r^* = \sum_{k=1}^{3} \left( \overset{\rightarrow k}{v}_r \cdot \overset{\rightarrow *}{F}_k + \overset{\rightarrow k}{\omega}_r \cdot \overset{\rightarrow *}{T}_k \right) \tag{21}$$

where the following is the case.

$$\overset{\rightarrow *}{F}_k = -m_k \overset{\rightarrow}{a}_k \tag{22}$$

$$\overset{\rightarrow *}{T}_k = -\overset{\leftrightarrow}{I}_k \cdot \overset{\rightarrow}{\alpha}_k - \overset{\rightarrow}{\omega}_k \times \overset{\leftrightarrow}{I}_k \cdot \overset{\rightarrow}{\omega}_k \tag{23}$$

In the above equations, $m_k$ is the mass of the $k^{\text{th}}$ rigid body, $\overset{\rightarrow}{a}_k$ is the acceleration of the mass center of the $k^{\text{th}}$ rigid body, $\overset{\rightarrow}{\omega}_k$ and $\overset{\rightarrow}{\alpha}_k$ are the angular velocity and angular acceleration of the $k^{\text{th}}$ rigid body, and $\overset{\leftrightarrow}{I}_k$ is the inertia dyadic of the $k^{\text{th}}$ rigid body about its mass center. The inertia dyadics of the three rigid bodies (tub, shaft, and drum) about their mass centers can be expressed as follows:

$$\overset{\leftrightarrow}{I}_1 = I_{11}^T \hat{t}_1 \hat{t}_1 + I_{22}^T \hat{t}_2 \hat{t}_2 + I_{33}^T \hat{t}_3 \hat{t}_3 \tag{24}$$

$$\overset{\leftrightarrow}{I}_2 = I_{11}^S \hat{s}_1 \hat{s}_1 + I_{22}^S \hat{s}_2 \hat{s}_2 + I_{33}^S \hat{s}_3 \hat{s}_3 \tag{25}$$

$$\overset{\leftrightarrow}{I}_3 = I_{11}^D \hat{d}_1 \hat{d}_1 + I_{22}^D \hat{d}_2 \hat{d}_2 + I_{33}^D \hat{d}_3 \hat{d}_3 + I_{12}^D \hat{d}_1 \hat{d}_2 + I_{21}^D \hat{d}_2 \hat{d}_1 \tag{26}$$

where the products of inertia $I_{12}^D$ and $I_{21}^D$ originate from the unbalance mass attached to the drum. Table 2 shows the mass and mass center locations of the three rigid bodies, and Table 3 shows the moments and products of inertia of the three rigid bodies. In Table 2, $x$, $y$, and $z$ denote $\hat{c}_1$, $\hat{c}_2$, and $\hat{c}_3$, which measure numbers of the mass center relative to the origin point $O$, respectively. The unbalance mass is considered as a point mass (400 g) attached to the drum at a point that is 0.2 m above the drum center in $\hat{d}_2$ direction.

**Table 2.** Mass and mass center locations of the three rigid bodies.

| Body | Mass (kg) | $x$ (mm) | $y$ (mm) | $z$ (mm) |
| --- | --- | --- | --- | --- |
| Body T | 23.87 | 240.61 | 0 | 0 |
| Body S | 4.34 | 18.11 | 0 | 0 |
| Body D | 4.04 | 209.9 | 19.8 | 0 |

**Table 3.** Moments and products of inertia of the three bodies.

| Body | $I_{11}$ (kg·mm²) | $I_{22}$ (kg·mm²) | $I_{33}$ (kg·mm²) | $I_{12} = I_{21}$ (kg·mm²) |
| --- | --- | --- | --- | --- |
| Body T | 1,030,000 | 1,610,000 | 1,530,000 | 0 |
| Body S | 55,300 | 38,400 | 38,400 | 0 |
| Body D | 196,000 | 162,000 | 177,000 | −7207 |

Forces acting on the washing machine are classified into conservative and non-conservative forces. Forces from the springs and bearings, the bending stiffness of the drum-shaft assembly, and gravitation are conservative, while those from dampers are non-conservative. Generalized active forces related to the conservative forces can be conveniently obtained by differentiating the potential energy of the system with respect to generalized coordinates. Figure 4 shows a spring and damper that connect the tub and cabinet. Table 4 shows the initial end point locations of the four springs and two dampers relative to the origin.

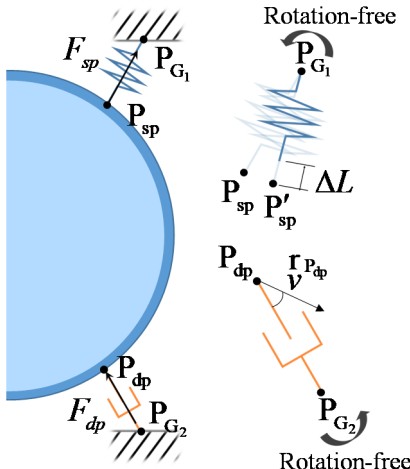

**Figure 4.** Spring and damper connecting the tub and cabinet.

**Table 4.** Initial end point locations of 4 springs and 2 dampers relative to the origin.

|  | Point Fixed to Cabinet | Point Fixed to Tub |
| --- | --- | --- |
| Spring 1 | (326.8, 300.2, 250.6) | (326.8, 162.6, 162.6) |
| Spring 2 | (326.8, 300.2, −250.6) | (326.8, 162.6, −162.6) |
| Spring 3 | (86.8, 300.2, 250.6) | (86.8, 162.6, 162.6) |
| Spring 4 | (86.8, 300.2, −250.6) | (86.8, 162.6, −162.6) |
| Damper 1 | (306.8, −280.0, 280.0) | (306.8, −162.6, 162.6) |
| Damper 2 | (306.8, −280.0, −280.0) | (306.8, −162.6, −162.6) |

Figure 5a shows the translational displacement measure between the front bearing centers: $(P_{fb})_T$ is attached to body T, and $(P_{fb})_S$ is attached to body S. Figure 5b shows the angular displacement measure $\phi$ between body D and body S about the axis $\hat{d}_3$ (or $\hat{s}_3$). The angular displacement measure can be obtained by using the cross product of $\hat{s}_1$ and $\hat{d}_1$. The generalized active forces $F_r$ consists of two components as follows:

$$F_r = (F_r)_C + (F_r)_N \quad (r = 1, 2, \cdots, 10) \tag{27}$$

where the following is the case:

$$(F_r)_C = -\frac{\partial U_C}{\partial q_r} \qquad (r = 1, 2, \cdots, 10) \tag{28}$$

$$U_C = U_E + U_G \tag{29}$$

$$U_E = \frac{1}{2}\left[\sum_{k=1}^{4} K_k(\Delta L_k)^2 + K_D\phi^2 + K_{fb}r_{fb}^2 + K_{rb}r_{rb}^2\right] \tag{30}$$

$$U_G = \sum_{k=1}^{3} m_k g h_k \tag{31}$$

$$(F_r)_N = \sum_{k=1}^{2} \vec{v}_r^{P_{dP}^k} \cdot \vec{F}_{dP}^k \quad (r = 1, 2, \cdots, 10) \tag{32}$$

$$\vec{F}_{dP}^k = -(C_{dp}\vec{v}^{P_{dP}^k} \cdot \frac{\vec{r}_{dP}^k}{\left|\vec{r}_{dP}^k\right|}) \frac{\vec{r}_{dP}^k}{\left|\vec{r}_{dP}^k\right|} \tag{33}$$

and where the following is obtained.

$$\vec{r}_{dP}^k = \vec{r}^{P_{G_k} P_{dP}^k} \tag{34}$$

$$\vec{v}^{P_{dP}^k} = \vec{v}^{P_D} + \vec{\omega}^D \times \vec{r}^{P_D P_{dP}^k} \tag{35}$$

$$\vec{v}_r^{P_{dP}^k} = \frac{\partial \vec{v}^{P_{dP}^k}}{\partial u_r} \tag{36}$$

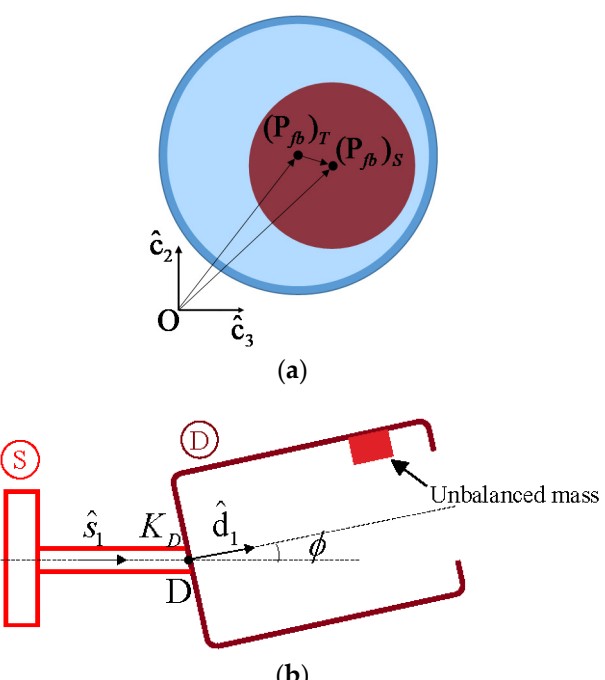

(a)

(b)

**Figure 5.** Relative translational and angular displacements (**a**) Relative displacement between bearing centers and (**b**) Relative angular displacement between body S and body D.

Notations $(F_r)_C$ and $(F_r)_N$ denote the generalized active forces obtained by the conservative and non-conservative forces, respectively. Potential energy $U_C$ is the sum of the elastic potential energy $U_E$ and gravitational potential energy $U_G$ of the three rigid bodies. Notations $K_k$, $K_{fb}$, $K_{rb}$, and $K_D$ are the stiffness values of the $k^{th}$ spring, front bearing, rear bearing, and drum-shaft assembly flexibility, respectively. Notations $\Delta L_k$, $r_{fb}$, $r_{rb}$, and $\phi$ are the translational and rotational deformation measures corresponding to the four stiffness elements. In Equation (31), $h_k$ is the $\hat{c}_2$ distance measure of the mass center of the $k^{th}$ rigid body relative to origin point $O$. Notation $C_{dp}$ is the damping constant of a damper. Notation $\vec{r}_{dP}^k$ is the vector defined by the two points of the $k^{th}$ damper shown in Figure 4. The initial locations of the points are given in Table 4. Using the generalized forces obtained in Equations (21) and (27), the equations of motion of the idealized washing machine model can be obtained as follows.

$$F_r^* + F_r = 0 \quad (r = 1, 2, \cdots, 10) \tag{37}$$

### 2.3. Linearization of the Equations of Motion

In order to efficiently obtain the linearized equations of motion of the idealized washing machine model, we need to carefully consider two things. First, premature linearization should be avoided. In other words, we should not linearize the velocities and angular velocities until we obtain linearized partial velocities and linearized partial angular velocities. Once we obtain those elements, we can linearize the velocities and angular velocities, with which we can obtain the linearized accelerations and angular accelerations. After we obtain the linearized partial velocities and linearized partial angular velocities in Equations (19) and (20), we can linearize the angular velocities and velocities in Equations (4)–(6) and Equations (8)–(10). Then, using the linearized angular velocities and velocities, we can obtain the linearized angular accelerations and accelerations. Finally, the linearized generalized inertia forces $\overline{F}_r^*$ can be obtained as follows:

$$\overline{F}_r^* = Linearize\left[\sum_{k=1}^{3}\left(\widetilde{v}_r^k \cdot \widetilde{F}_k^* + \widetilde{\omega}_r^k \cdot \widetilde{T}_k^*\right)\right] \tag{38}$$

where $\widetilde{v}_r^k$ and $\widetilde{\omega}_r^k$ denote the linearized partial velocities and linearized partial angular velocities, and $\widetilde{F}_k^*$ and $\widetilde{T}_k^*$ denote the linearized inertia forces and linearized inertia torques, which can be given as follows:

$$\widetilde{F}_k^* = -m_k\widetilde{a}_k \tag{39}$$

$$\widetilde{T}_k^* = -\overset{\leftrightarrow}{I}_k \cdot \widetilde{\alpha}_k - \widetilde{\omega}_k \times \overset{\leftrightarrow}{I}_k \cdot \widetilde{\omega}_k \tag{40}$$

where $\widetilde{a}_k$, $\widetilde{\alpha}_k$, and $\widetilde{\omega}_k$ denote linearized acceleration, linearized angular acceleration, and linearized angular velocity.

Second, the generalized active forces should be linearized around the static equilibrium of the washing machine. Thus, a first-order Taylor series expansion should be obtained for the linearization of the nonlinear spring and damping forces. Then, the final linearized equations of motions can be expressed by using a matrix form as follows:

$$\mathbf{M}\ddot{\mathbf{q}} + \mathbf{C}\dot{\mathbf{q}} + \mathbf{K}\mathbf{q} = \mathbf{F} \tag{41}$$

where M, **C**, and **K** are $10 \times 10$ square matrices, and **F** is a $10 \times 1$ column matrix. If the rotational speed of the drum is constant, some elements of the mass and stiffness matrices M and **K** are given as harmonic functions of time. However, since the unbalance mass is very small compared to the tub and drum masses, the harmonic terms hardly affect the modal characteristics of the total system. The harmonic terms that belong to **F**, however, significantly affected the transient characteristics of the washing machine.

## 3. Numerical Results and Discussion

### 3.1. Validation of the Nonlinear Analytical Model

The nonlinear equations of motion derived in Section 2 were numerically solved using MATLAB, and the results were compared with those obtained with the commercial multibody dynamic analysis software RecurDyn, v. 9R1 [10]. To obtain the results, static equilibrium analysis was carried out first before the transient analysis began. Thus, the displacements shown in Figure 6 are the values obtained from the static equilibrium. The stiffness coefficient of a spring is given as $K_k = 8000\,\text{N/m}$, and the damping constant of a damper is $C_{dp} = 140\,\text{N} \cdot \text{s/m}$. The bending stiffness of the drum-shaft assembly is given as $K_D = 29670\,\text{N} \cdot \text{m/rad}$, and the radial stiffness of the bearing is given as $K_{fb} = K_{rb} = 2 \times 10^8\,\text{N/m}$. Starting from 0 rpm, operation speed increased up to 1200 rpm in 20 s and remained at the final angular speed until 30 s. Therefore, the prescribed rotational speed of the drum is given as a function of time as follows.

$$\begin{aligned} \dot{\theta}(t) &= 40\pi\left[\tfrac{t}{20} - \tfrac{1}{2\pi}\sin\left(2\pi\tfrac{t}{20}\right)\right] && 0 \leq t \leq 20 \\ \dot{\theta}(t) &= 40\pi && t > 20 \end{aligned} \tag{42}$$

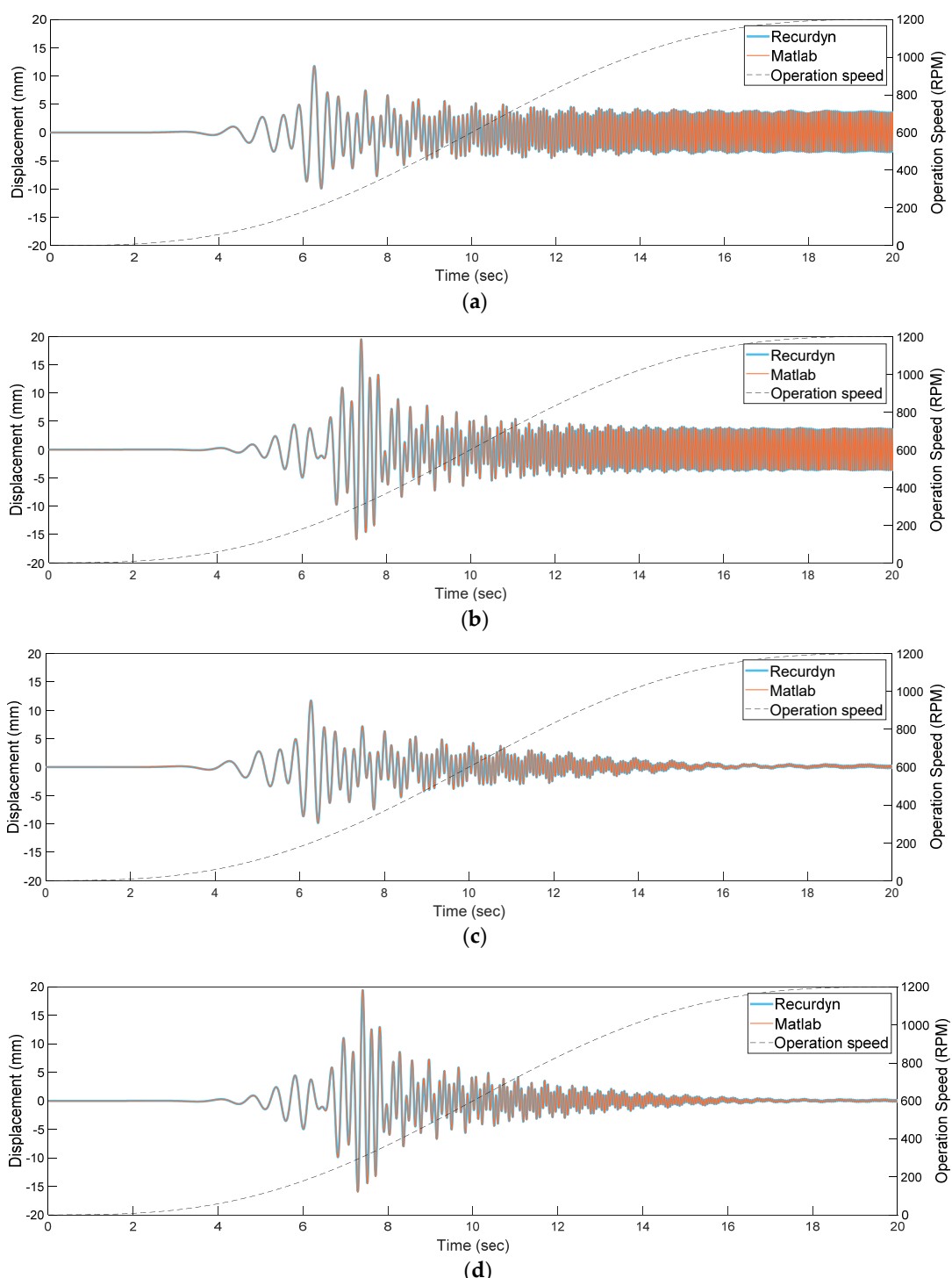

**Figure 6.** Comparison of dynamic analysis results obtained with the nonlinear analytical model and commercial software: (**a**) tub front center horizontal displacement, (**b**) tub front center vertical displacement, (**c**) drum front center horizontal displacement, and (**d**) drum front center vertical displacement.

Figure 6 shows the horizontal and vertical displacements of the front center point $P_{TE}$ of the tub and the front center point $P_{DE}$ of the drum from the equilibrium position, which can be obtained by using equilibrium analysis. The figure shows that the transient analysis results obtained with the nonlinear analytical model derived in this study are in good agreement with those obtained with the RecurDyn software.

### 3.2. Modal Analysis with the Linear Model

Modal analysis was conducted using Equation (41) obtained in the previous section. A complex modal analysis method was employed by using the state-space formulation. Since the drum rotates with the unbalance mass, some elements of the mass and stiffness matrices in Equation (41) change periodically; thus, the linear dynamic model is not autonomous. Given those periodic changes, the maximum variation of the natural frequencies due to the rotating unbalance is less than 0.3% of the mean values of the natural frequencies throughout the operation speed range (0 to 1200 rpm). The results are shown in Figure 7b. Thus, we may ignore the variation effect due to the rotating mass unbalance when we solve the following complex eigenvalue problem:

$$\{s\mathbf{M}^* + \mathbf{K}^*\}\psi = 0 \tag{43}$$

where the following is the case.

$$\mathbf{M}^* \equiv \begin{bmatrix} 0 & \mathbf{M} \\ \mathbf{M} & \mathbf{C} \end{bmatrix} \tag{44}$$

$$\mathbf{K}^* \equiv \begin{bmatrix} -\mathbf{M} & 0 \\ 0 & \mathbf{K} \end{bmatrix} \tag{45}$$

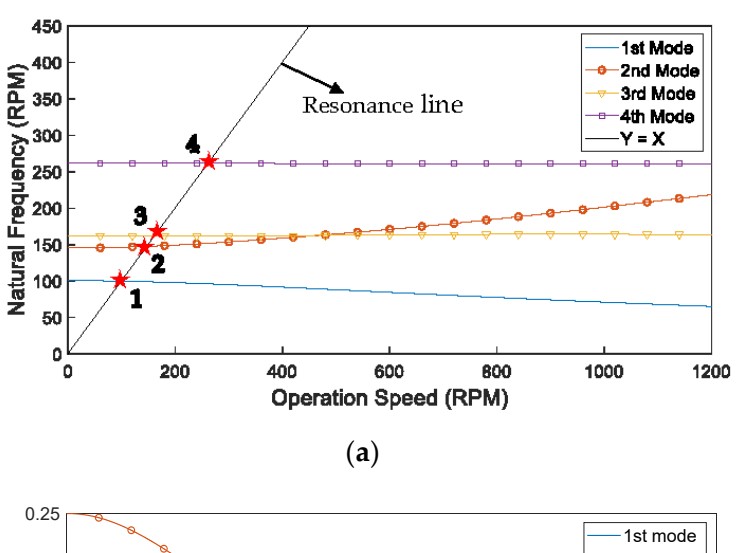

(**a**)

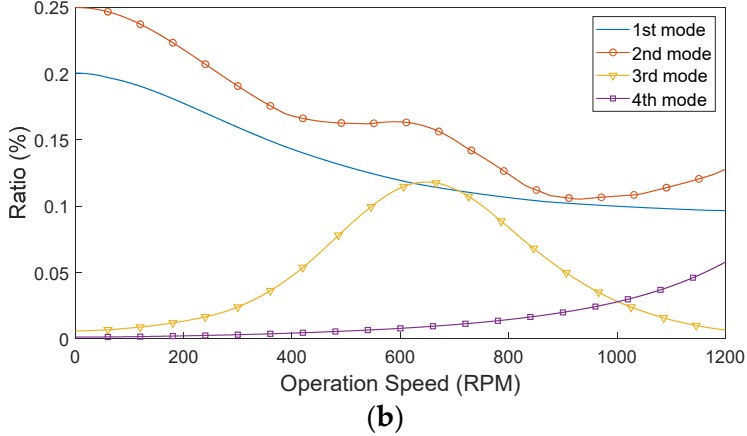

(**b**)

**Figure 7.** Lowest four natural frequencies and their maximum variations versus operation speed: (**a**) lowest four natural frequencies with resonance line and (**b**) maximum variations of the lowest four natural frequencies.

By solving the eigenvalue problem shown in Equation (43), the complex eigenvalue *s* and the eigenvector **ψ** can be obtained. The lowest four natural frequencies versus the

operating speed are shown in Figure 7a. Resonance might occur when the operating speed coincides with the natural frequencies. Four possible resonance speeds are marked with red stars in the figure.

Four mode shapes of the idealized washing machine model at the four resonance speeds are shown in Figure 8. The first resonance mode is a rocking motion in the horizontal plane at 99.9 rpm, and the second resonance mode is a rocking motion in the vertical plane at 147.6 rpm. The third resonance mode is a translational motion in the horizontal plane at 162.3 rpm, and the fourth resonance mode is a translational motion in the vertical plane at 261.6 rpm. These four resonance modes could be also exhibited by transient analyses with the four resonance speeds using a nonlinear analytical model or the commercial software RecurDyn.

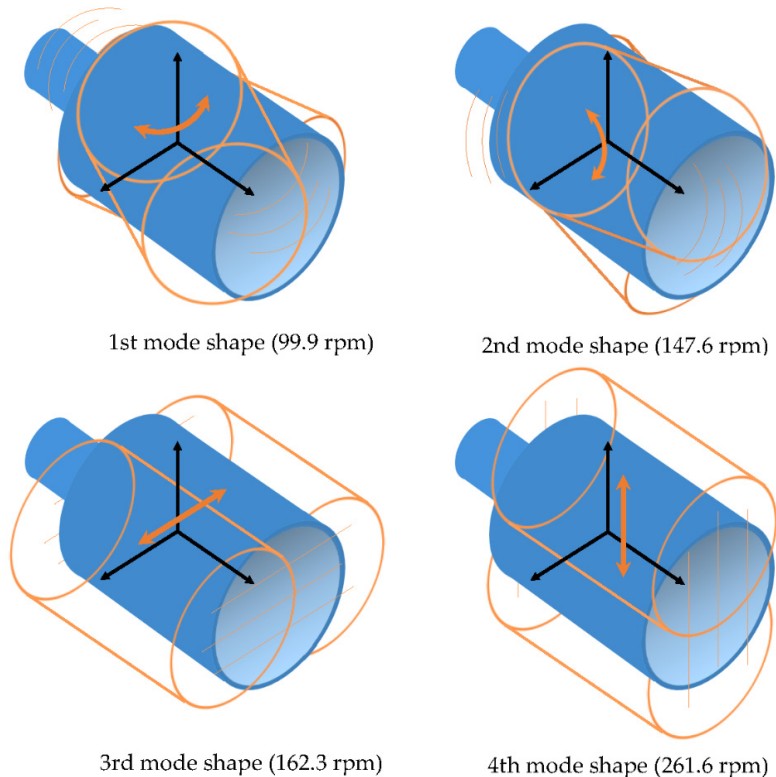

1st mode shape (99.9 rpm)          2nd mode shape (147.6 rpm)

3rd mode shape (162.3 rpm)          4th mode shape (261.6 rpm)

**Figure 8.** Lowest four resonance mode shapes of the front-loading type washing machine model.

*3.3. Comparison of the Nonlinear and Linear Analytical Models*

Figure 9 shows the transient responses obtained with the nonlinear and linear analytical models. The simulation condition is same as the one provided in Section 3.1. As shown in Figure 9, transient responses of the linear analytical model are in good agreement with those obtained with the nonlinear analytical model. The gray vertical lines denote the times when the operation speed passes the four resonance speeds. The first and third mode shapes are related to the vertical movement of the drum front center, and the second and fourth mode shapes are related to the horizontal movement of the drum front center. In the transient responses, the transient response amplitude peaks do not exactly match with the resonance speeds. The amplitude peaks occur after the operation speed passes the resonance speeds; these phenomena of resonance delay were discussed in previous studies [11,12]. The figure shows that the third and fourth translational modes are more critical for tub vibration during the transient state than are the first and second rocking modes. The variation of the minimum gap size between the tub and the drum during the operation is also shown in Figure 9c. This shows that a clash between the tub and the drum does not occur for the washing machine during operation.

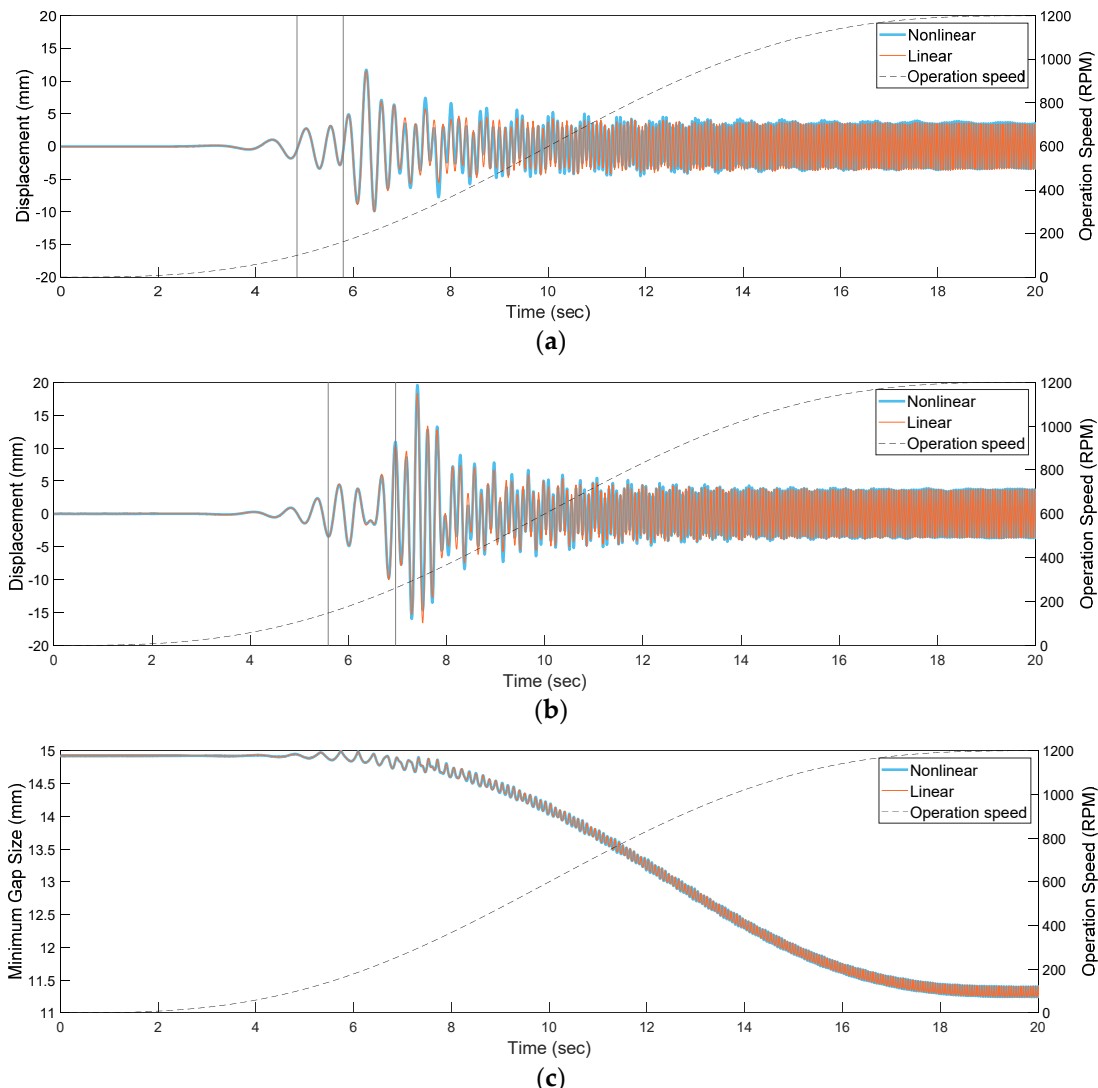

**Figure 9.** Comparison of transient responses obtained with the nonlinear and linear analytical models: (**a**) tub front center horizontal displacement, (**b**) tub front center vertical displacement, and (**c**) minimum gap size between the tub and drum.

Figure 10 shows the comparison of transient responses obtained with the nonlinear and linear analytical models when the damping constant $C_{dp}$ is 140 N·s/m. The operation speed increases and decreases linearly in the range of 0~350 rpm to determine whether the previously obtained resonance mode shapes actually appear in the transient responses. Again, the gray vertical lines indicate the times when the operation speed passes through resonance speeds. The horizontal and vertical displacements of the mass center corresponding to the third and fourth translational resonance modes are shown in Figure 10a,b; the rocking angles in the horizontal and vertical plane corresponding to the first and second rocking modes are shown in Figure 10c,d. Transient peaks can be observed after the operation speed passes through the resonance speeds obtained in Figure 7. In other words, resonance mode shapes and resonance speeds obtained from modal analysis are useful for predicting the transient characteristics of the washing machine. In Figure 10c,d, we can observe that the rocking angles obtained with the nonlinear analytical model are somewhat different from those obtained with the linear analytical model. The difference appears when the angular speed increases, but it almost disappears when the angular speed decreases. This is a typical phenomenon that frequently occurs in a nonlinear vibration system. Such a phenomenon, often called the jump phenomenon (see [13]), cannot be predicted with a linear analytical model.

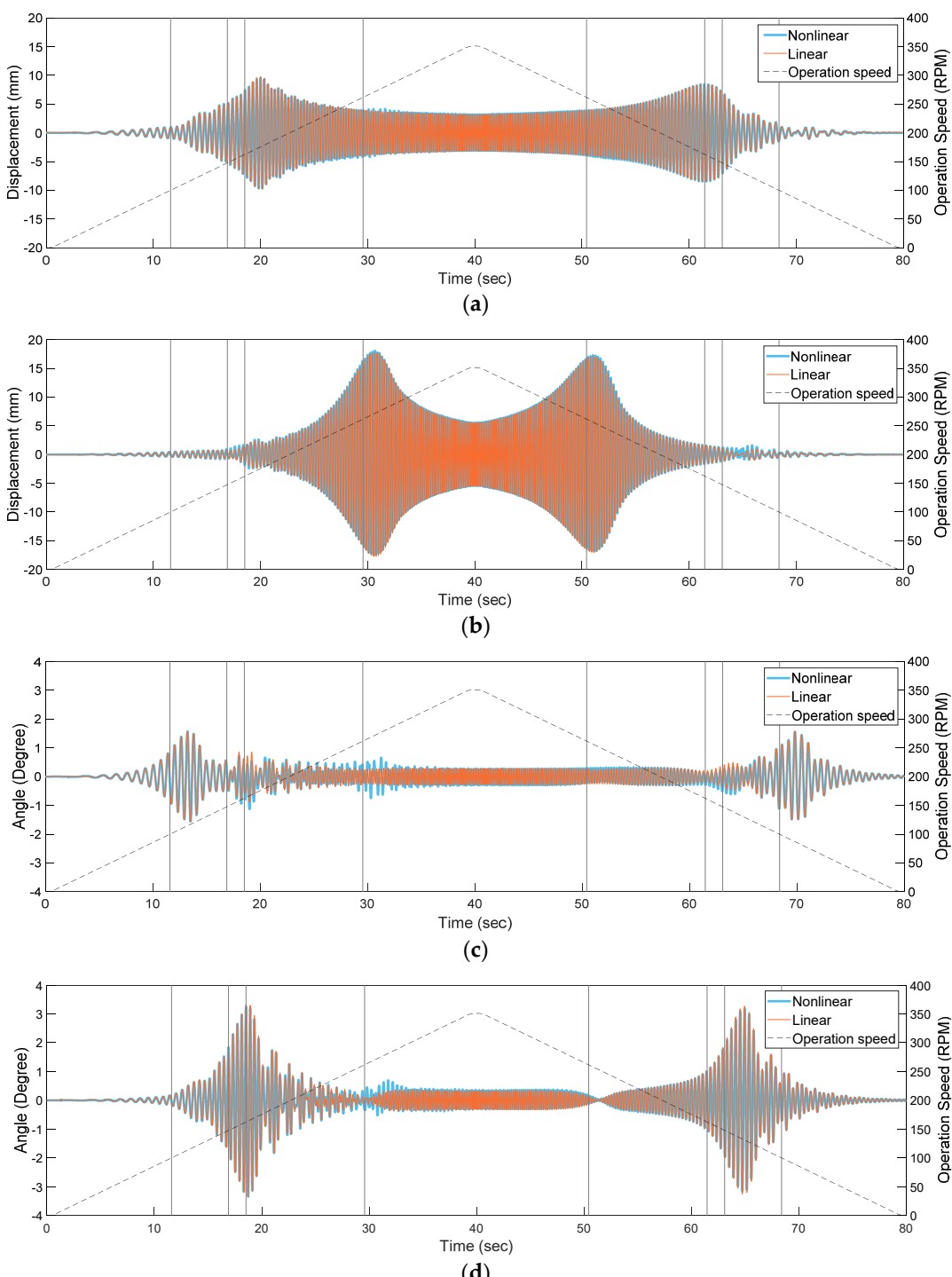

**Figure 10.** Comparison of transient analysis results obtained with the nonlinear and linear analytical models (when = 140 N · s/m): (**a**) horizontal displacement of the mass center, (**b**) vertical displacement of the mass center, (**c**) rocking angle in the horizontal plane, and (**d**) rocking angle in the vertical plane.

Figure 11 shows the rocking angles in the horizontal and vertical planes obtained with the nonlinear and linear analytical models when the damping constant $C_{dp}$ is reduced to 100 N · s/m, which is smaller than the previously used reference value 140 N · s/m. Significant rocking motion in the horizontal and vertical planes can be observed with the nonlinear analytical model after operation speed passes the fourth resonance speed. With the linear analytical model, however, such violent rocking behavior cannot be predicted.

This indicates that the reliability of the linear analytical model depends on the damping constant employed for the model. The results shown in Figures 10 and 11 indicate that the reliability of the linear model can be guaranteed with a sufficiently large damping constant value.

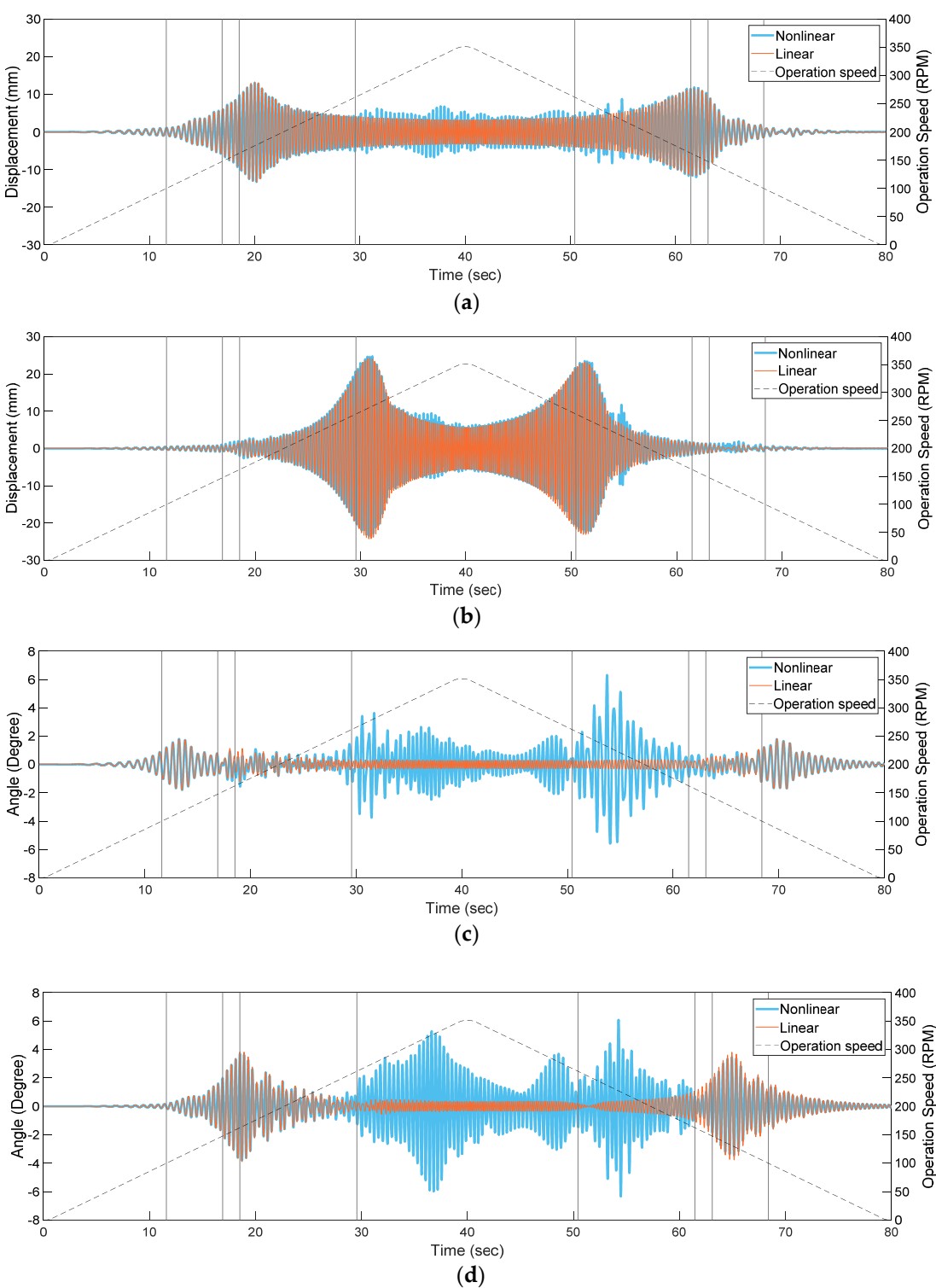

**Figure 11.** Comparison of the rocking angles obtained with the nonlinear and linear analytical models (when = 100 N · s/m): (**a**) horizontal displacement of the mass center, (**b**) vertical displacement of the mass center, (**c**) rocking angle in the horizontal plane, and (**d**) rocking angle in the vertical plane.

Figure 12 shows the parameter range to guarantee the reliability of the linear analytical model. We obtained this reliability diagram by comparing the transient response amplitudes obtained with the linear and nonlinear analytical models in the time range of 30–50 s When the maximum amplitude ratio (the transient response amplitude obtained with the nonlinear analytical model to that obtained with the linear analytical model) in the time range is smaller than two, the linear analytical model is considered reliable; otherwise, it is considered unreliable. Therefore, the linear analytical model can be effectively used for the design of an FL type washing machine when the unbalance mass and damping parameters are chosen from the reliable range.

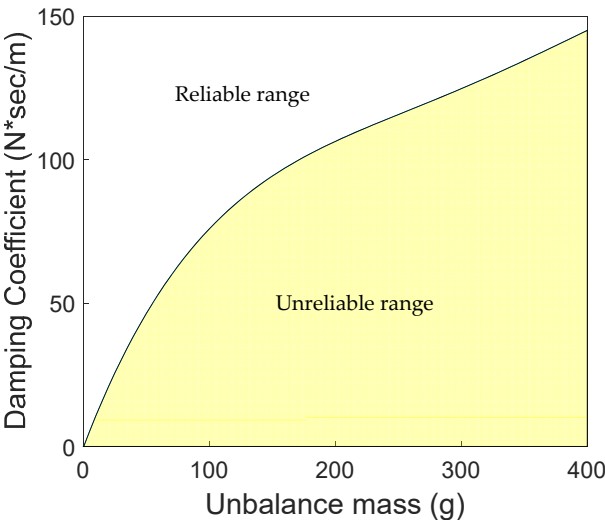

**Figure 12.** Reliability diagram of the linear analytical model.

### 4. Conclusions

In order to investigate the vibration characteristics of an FL type washing machine, a dynamic model consisting of three rigid bodies, revolute joints, linear springs, linear dampers, and stiffness elements such as bearings and flexible hinges was introduced, and the equations of motion of the idealized model were derived using Kane's method. The accuracy of the nonlinear analytical model was first validated by comparing the numerical results with those obtained with the RecurDyn software. The nonlinear analytical model was then linearized around the equilibrium position in order to obtain the linear analytical model. Since the rotational motion of the drum was prescribed as a function of time, the linear analytical model was a non-autonomous system. However, modal analysis could be carried out effectively with the linear analytical model since the variations of the natural frequencies due to the time-varying terms in the linear analytical model were trivial. We could predict the resonance mode shapes with the linear model, and the transient characteristics of the linear analytical model could be validated with the transient responses obtained at the resonance speeds.

The parameter study showed that the reliability of the linear analytical model could be guaranteed if the damping constant was larger than a certain value when a rotating unbalance mass was given. When the damping constant was not sufficiently large, however, we could not obtain reliable transient responses with the linear analytical model. As the rotating unbalance mass increases, the damping constant should be increased to guarantee the reliability of the linear analytical model. Therefore, the reliability range related to the unbalance mass and the damping constant should be accurately identified before the linear analytical model is used for the design or control of an FL type washing machine.

**Author Contributions:** J.P. is the principal investigator of the research described in this manuscript. He developed the analysis methodology and performed the validation work. S.J. and H.Y. are the corresponding authors of this manuscript. They participated in the writing, review, and editing

of the manuscript and provided supervision for the research. H.Y. was responsible for the project administration and funding acquisition. All authors have read and agreed to the published version of the manuscript.

**Funding:** This research was funded by National Research Foundation of Korea (NRF) funded by the Ministry of Science, ICT, grant number NRF-2018R1A2A2A05022590.

**Institutional Review Board Statement:** Not applicable.

**Informed Consent Statement:** Not applicable.

**Conflicts of Interest:** The authors declare no conflict of interest. The funders had no role in the design of the study; in the collection, analyses, or interpretation of data; in the writing of the manuscript; or in the decision to publish the results.

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
