# Peer review of "Dynamic Modeling of a Front-Loading Type Washing Machine and Model Reliability Investigation"

_machines, doi:10.3390/machines9110289_

Round 1

Reviewer 1 Report

  • The resolution of figures are not clear to figure out the correlations which are interesting in this research, especially figures 6, 9, 10
  • Check Table number placement, top or bottom? 
  • Section 2.3 for linearization is too short to explain, some more better!
  • Check numbering format for EQs 38, 29, 40
  • If you have experimental results, the paper could have more credit, but it is not mandatory.        

Reviewer 2 Report

Modeling of the FL-type washing machine and the reliability analysis of the linearized model have been discussed in this paper.  The reliability analysis is important to understand the effectiveness of the linear model in the design phase.  

However if authors improve the following points, the paper would be more valuable in this Journal.

  • To enhance the paper, the reviewer recommand that the  authors mention that the three body model is good enough to characterize important transient dynamic effects by referring previous studies.
  • In section 2.3, derivation of the linearized equations of motion has been abstractly summarized in Eq. (38). Recommend that authors provide more explanation (or explicit expressions) about the difference between the linearized acceleration, the linearized angular accelerations and the linearized angular velocities in Eqs (39) and (40) by comparing to those in Eqs (22), and (23).
  • Like the Figure 10, the translational displacements for horizontal and vertical motion must be also included in Figure 11 in order to readers can understand the resonance effect due to the reduction of the damping parameter.
  • The visibility of the fugures are so poor. All the figures must be replaced with high resolution figures.

Followings are the minor typo graphical errors.

  • Page 2, ---- The accuracy of the nonlinear analyt[1]ical model was first validated by comparing its numerical results to those obtained with a commercial multibody dynamic analysis software [10].

Reference 10 should be changed to the Ref 13.

  • Equations numbers (21)-(23) and (38)-(40) must be located in the right end.
  • The caption of the table 2 must be located before the actual table.

Reviewer 3 Report

In this paper, a linear dynamic model of a front-loading type washing machine is developed. Based on the Kane’s method, the fully nonlinear equations of motion of the idealized washing machine composed of three rigid bodies are derived. Then, by linearizing the nonlinear equations, the linear equations of motion are obtained and the reliability of the linear dynamic model is investigated. The parameters relevant to the reliability of the linear dynamic model are identified and the reliability ranges of the proposed linear dynamic model are numerically investigated. The paper is clearly written, and the conclusions are useful in the design of the front-loading type washing machine. The paper is suggested to be modified according to the following comments: 1. The formats of the equations should be the same, for example, the number of each equation is located at the right side of the page. 2. Page 7, the title of the table 2 should be above the table 2. 3. Page 8, the title of the table 3 should be close to the table 3. 4. The definition of s6, c6, s8, c8 is suggested to be added after equation (6). 5. Figure 6, 9 and 10 are not clear, which should be replaced. 6. It is said in page 16 that when the maximum amplitude ratio in the time range is smaller than 2, the linear analytical model is considered reliable. It seems that the accuracy of the linear analytical model is not sufficient. It is suggested to use the relative error for comparison.

Reviewer 4 Report

The authors present a comparison between nonlinear and linear analytical dynamical models of a front-loading washing machine operating at rotational speeds between 0 and 1200 rpm, making an assessment of the reliability and usefulness of the linear approximation depending on the damping and the magnitude of the unbalanced mass inside the drum.

Formally, the paper is well written, but the readability of some of the figures is not good; in general, the definition of the figures should be increased.

Regarding its scientific content, the subject is interesting but the adopted methodology arises some doubts.   In particular, I have the following comments:

- The paper does not quantifies the advantages of using a linear model instead of the nonlinear one.   Because he model is small (10 dofs), the computational cost of solving the nonlinear model should not be very large. Is it really worth to work with the linear model, with its limitations?

- The transient responses depicted in Figure 6 start from 0. Does it mean that the static position is calculated previously, and the figure shows the displacement from this position?

- Section 3.2. uses a state-space approach for the modal analysis. Please, provide a brief justification for it.

- In section 3.2 it is stated that the maximum variation of the natural frequencies due to the unbalanced mass is less than 0.3% of the mean value, but Figure 7a) shows that the frequency of mode 2 changes significantly. What is more, if Figure 7b) shows the variations from the mean value, I understand that for some intermediate speed the variation should be zero.

- In section 3.3, line 5, the description of the modes is not correct; first and third are horizontal and second and fourth are vertical.

- Page 15, 5 lines from the end of the page, it is stated “This is a typical phenomenon that frequently occurs in a nonlinear vibration system”. Can you provide a reference for this?

- In sections Abstract and Conclusions is stated that the reliability of the linear analytical model could be guaranteed for a range of damping and mass. I think that this conclusion is too general.   The reliability diagram in Figure 12 is valid only up to 350 rpm.  What happen for higher speeds, such as the ones considered in Figures 6 and 9?
